# Vaccination as a Strategy to Prevent Bluetongue Virus Vertical Transmission

**DOI:** 10.3390/pathogens10111528

**Published:** 2021-11-22

**Authors:** José M. Rojas, Verónica Martín, Noemí Sevilla

**Affiliations:** Centro de Investigación en Sanidad Animal, Instituto Nacional de Investigación y Tecnología Agraria y Alimentaria, Consejo Superior de Investigaciones Científicas (CISA-INIA-CSIC), 28130 Valdeolmos, Spain; veronica.martin@inia.es

**Keywords:** orbivirus, bluetongue, vaccination, vertical transmission, review

## Abstract

Bluetongue virus (BTV) produces an economically important disease in ruminants of compulsory notification to the OIE. BTV is typically transmitted by the bite of *Culicoides* spp., however, some BTV strains can be transmitted vertically, and this is associated with fetus malformations and abortions. The viral factors associated with the virus potency to cross the placental barrier are not well defined. The potency of vertical transmission is retained and sometimes even increased in live attenuated BTV vaccine strains. Because BTV possesses a segmented genome, the possibility of reassortment of vaccination strains with wild-type virus could even favor the transmission of this phenotype. In the present review, we will describe the non-vector-based BTV infection routes and discuss the experimental vaccination strategies that offer advantages over this drawback of some live attenuated BTV vaccines.

## 1. Introduction

Bluetongue (BT) is a disease of mandatory notification to the World Organization of Animal Health (OIE) that causes important economic losses globally estimated to be around three billion dollars per year [1]. Bluetongue virus (BTV) is the etiological agent responsible for BT, a disease that affects domestic and wild ruminants and that can be particularly severe in sheep [2,3]. BT disease clinical signs are characterized by the virus preferred tropism for endothelial cells [4]. As a consequence of endothelial cell damage, edema and hemorrhages can take place in BTV infections. Early clinical signs are pyrexia, depression, and loss of appetite [5,6]. In some cases, the disease progresses to conjunctivitis, congestion of the nasal and oral mucosa and edema of the face and lip. Sometimes hemorrhagic lesions occur which can progress to the cyanosis of the tongue that gave its name to the disease. In the most severe cases, respiratory distress and esophageal paresis can develop which can ultimately lead to the death of the infected animal. Although BTV infection is not always fatal, it typically leads to reduced productivity in ruminants (e.g., reduced milk yield, weakness of the animal, abortion or stillbirth)[5,6]. BTV therefore produces a debilitating disease that affects the livestock industry.

BTV circulation was once restricted to the subtropical regions with occasional incursion in more temperate areas of the globe. However, it has now become apparent that the disease has become endemic in the European part of the Mediterranean basin [7,8]. Vaccination can control outbreaks; however, at least 28 different BTV serotypes with little cross-reactivity have been identified so far [9,10,11,12]. This complicates disease control as multiple vaccines are required for protection in regions where multiple serotypes are circulating. BTV serotypes have been classified as “classical” (serotypes 1–24) or “atypical” for some recent isolates that predominantly affect small ruminants with little to no clinical signs [13,14,15,16]. Only “classical” BTV serotypes (1–24) are notifiable to the OIE [17].

## 2. BTV Viral Particle

BTV belongs to the *Reoviridae* family and is prototypical of the *Orbivirus* genus. BTV is a double stranded RNA (dsRNA) virus, its genetic material consists of 10 segments (Figure 1) [18], encoding for 7 structural proteins (VP1 to VP7) and at least 4 non-structural proteins (NS1 to NS4). A putative fifth non-structural protein (NS5) has also been reported [19]. The viral particle consists of a two-layer core that encapsulates the RNA polymerase and the segmented genome. The outer core is composed of the highly variable VP2 protein and the VP5 protein that acts as the main anchor of this layer to the inner core. This outer core is responsible for the interaction with the host cellular components that allow virus cell entry. Most neutralizing antibodies are also directed against the proteins in this layer and mostly against VP2. The high variability of VP2 confers the virus with a means to evade neutralizing antibodies, which, as a result, generates the 28 serotypes with little cross-reactivity.

Once internalized, the outer core is destabilized by low pH, which allows VP5-mediated liberation of the highly stable inner core into the cytoplasm [20,21]. The inner core is composed of the VP7 and VP3 proteins and serves as a protective shell for the viral replication machinery. VP3 also anchors the RNA polymerase VP1 to the inner core [22]. Core-like particle assembly experiments have also indicated that the RNA capping enzyme and methyl transferase VP4 is associated with the VP3-VP1 complex [23]. The spatial distribution within the core of the RNA helicase VP6 is less well characterized, but its presence is important for the correct packaging of the dsRNA genome [24]. The inner core also contains the segmented RNA genome.

Non-structural proteins are involved in promoting viral replication in the host cells and in interfering with immunity. NS1 forms cytoplasmic tubules that promote viral protein expression [25]. NS1 enhancement of viral mRNA translation relies on two zinc finger-like motifs present in the protein and on the transition from the inactive tubular state to an active non-tubular form [26]. NS2 is the most abundant protein in viral inclusion bodies (VIB). VIB formation is dependent on NS2 phosphorylation, which is enhanced by calcium ions [27]. NS2 is an RNA binding protein that facilitates the assembly of new viral inner cores [28]. NS3, and its shorter isoform NS3a which lacks the first 13 N-terminal amino-acid residues, is involved in virion egress [29,30,31]. NS3 can act as a viroporin, thus easing the release of new viral particles [32]. NS3 also contributes to the maturation of the viral particle, possibly through its binding to VP2 [33] which promotes the release of two-layered mature viral particles. VP3, NS3, NS4 and the putative NS5 are involved in countering the antiviral cell response. VP3, NS3 and NS4 can act as IFN antagonists (reviewed in [34]). VP3 can impair IFN induction [35], while NS3 and NS4 can counter IFN induction as well as type I and type II IFN signaling [36,37,38,39,40]. Finally, the putative NS5 has been shown to promote cellular shut-off in transfection experiments [19].

## 3. BTV Is Mainly an Arbovirus, but It Can Be Transmitted through Other Routes

BTV is principally an arthropod-borne virus (arbovirus) that is transmitted by the bite of *Culicoides* spp. to ruminants [41] (Figure 2A). However, BTV can also be transmitted through other routes (Figure 2B–E). There is evidence that large African carnivores can become infected probably through feeding on BTV-infected ruminants [42]. Similarly, BTV-8 could be transmitted to Eurasian lynx through this oral route [43]. The significance of these findings in the wider context of BTV transmission is unclear, but it is unlikely to have a high epidemiological impact.

Horizontal transmission in ruminants of BTV-1, BTV-2 and BTV-8 has been documented under experimental conditions [44,45,46]. Naïve animals housed with infected counterparts can, in some cases, become infected. This transmission route is probably the result of animals being in close proximity and/or sharing food and water troughs. Oral transmission in ruminants is also suspected in the field as a result of ingestion of contaminated placenta or colostrum [47,48]. The direct contact route appears to be particularly important in the transmission of some “atypical” BTV serotypes that specifically infect small ruminants [49,50]. From an epidemiological perspective, horizontal transmission is unlikely to be a major component of epizootic episodes, although it could have an impact on BTV morbidity in farms with densely housed livestock.

Since BTV possesses an affinity for erythrocytes [51], it is plausible that it can be transmitted through mechanical means. Indeed, there are instances in which this transmission route has been demonstrated. Transmission through sharing infected needles is documented, even in the absence of visible blood contamination (subcutaneous inoculation) [52], thus indicating that sharing needles for inoculations between ruminants poses a risk of BTV transmission. Tick transmission has also been documented [53]. Indeed, several species of ticks can become infected by BTV-8, and the virus can be found in salivary glands and thus could potentially be transmitted to ruminant hosts [54]. In the same study, BTV was shown to pass transstadial stages in hard ticks (nymph to adult) and to infect eggs in soft ticks [54]. These phenomena could contribute to BTV overwintering mechanisms, although this has yet to be confirmed. Tick transmission is nonetheless unlikely to be a major route of disease spreading.

BTV is also known to target ram and bull semen quality and can be isolated from this fluid in infected animals [55,56]. Recently, BTV transmission to heifers through insemination with semen from naturally infected bulls has been demonstrated [57]. Previous reports had already established that BTV could be transmitted through this route using semen from experimentally infected ruminants [58,59]. This transmission route has implications in disease control, as it is now suspected that the re-emergence of BTV-8 in France in 2015 could be the result of insemination with frozen semen obtained from a 2008 infected animal [60].

Vertical transmission from the pregnant female to the fetus is the alternative BTV transmission route with the most epidemiological significance. Indeed, venereal BTV transmission can also result in vertical transmission of the virus to the fetus, which often leads to abortions [57]. Vertical BTV transmission was first suspected in the 1950s as a result of vaccination with a live attenuated virus that increased stillbirth and weak lambs in vaccinated flocks [61]. Transplacental transmission was subsequently confirmed in sheep, cattle, goat and elk [62,63,64,65,66]. For a while, transplacental transmission was associated with live attenuated vaccine strains that had been passaged in embryonated chicken eggs (expertly reviewed in [63]). However, this feature has now also been associated with some BTV field strains [44,46,64,67,68,69,70,71] such as the BTV-8 responsible for the 2006 European outbreak, and thus, vector infected ruminants can transmit the virus to their offspring. It should be noted that BTV vertical transmission depends greatly on isolates. The factors that govern BTV vertical transmission are unknown but appear to be intrinsic to the virus as the rescued reverse genetic virus of a BTV-2 strain known to cross the placental barrier maintained this phenotype [44]. Curiously, BTV effects on reproduction are not limited to the ruminant hosts of the disease. There is evidence that BTV can produce abortions in dogs and even cause mortality in pregnant bitches [72,73,74,75]. This further indicates that BTV possesses intrinsic mechanisms that allow it to cross the placental barrier. From an epidemiological perspective, vertical transmission could be involved in overwintering, as newborn calves/lambs can be BTV positive, and thus could potentially start a new cycle of infection by passing the virus to the arthropod vector. Thus, vertical transmission is an aspect of BT disease that needs close attention.

**Figure 2 pathogens-10-01528-f002:**
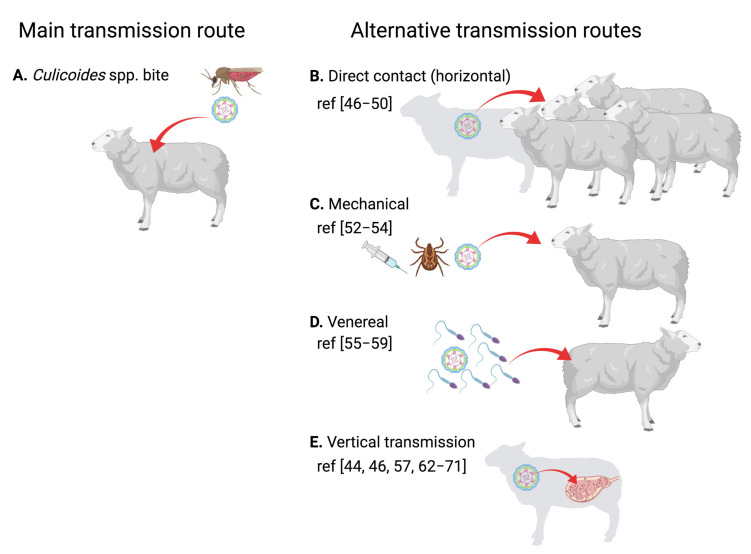
Transmission routes of BTV in ruminants. (**A**) Typically, BTV is transmitted to the mammalian host through the bite of infected *Culicoides* spp. (**B**–**E**) Other transmission routes have nonetheless been documented. (**B**) BTV can be transmitted by direct contact in some rare cases, probably through sharing of water and food trough or consumption of infected placenta or colostrum. (**C**) BTV affinity for erythrocytes makes mechanical transmission possible. Infection by sharing contaminated needles and transmission by tick bites has been documented. (**D**) Venereal transmission through the semen of infected ruminants has also been demonstrated. (**E**) Finally, vertical transmission from the mother to the fetus is associated with some BTV strains. This often leads to abortions, stillbirths or lambs/calves with neurological issues. (Created with Biorender.com).

## 4. Impact of BTV Vertical Transmission

As previously stated, BTV infection in pregnant cows and ewes can lead to abortion or weak offspring. This represents an important economic setback for livestock farming. Moreover, transplacental transmission could contribute to BTV overwintering mechanisms. This aspect of BTV infection is often underestimated, and indeed a study found a 56% probability of vertical transmission events for BTV-8 which indicates that this transmission route could be more frequent than previously thought for some BTV isolates [76].

Early studies established BTV tropism for brain tissue in infected fetuses that resulted in congenital brain malformation [77,78,79]. The structural protein VP5 has been associated with viral neural tropism in newborn mice [80]. The teratogenic effects on fetuses of BTV during gestation depend greatly on the time of infection (reviewed in [81]). Effects on the fetuses are more severe at the early stage of gestation, and they appear to decrease as fetus immunocompetence develops from days 60–70 in sheep and days 120–130 in cattle [81,82,83]. Nonetheless, brain affectations, such as encephalitis, can still be detected in animals apparently born healthy but that were exposed to the virus [84]. Vertical transmission appears to be more likely when infection occurs in early to mid-gestation [85,86,87].

Fetus exposure to BTV in early pregnancy leads to cavitating white matter brain lesions [77] that are the results of the destruction by the virus of stem cells from the central nervous system [81]. Once pregnancy advances and the BTV-susceptible glial and neuronal precursor cells migrate to the white matter, the teratogenic effects of BTV infection in fetuses are diminished [81]. Infections in late pregnancy typically produce mild encephalitis and premature births [84,88,89]. Newborn calves/lambs exposed to BTV in utero can be born PCR positive. This has been proposed as a mechanism for virus overwintering in climates in which vector activity is greatly reduced in winter [62]. Indeed, the virus can be isolated in some instances from newborn calves [67] and newborn calves can remain PCR positive for up to five months [70], which supports the idea that transplacental transmission can lead to BTV overwintering.

In most cases, newborns that became infected in utero develop antibodies and are seropositive at birth. In some cases, PCR positive but seronegative calves have been reported [67]. This could be indicative of a tolerance to BTV, which could lead to chronic infection in these animals. Given the differences between the infant and adult immune system, viral infection in early life can have very different outcomes to infection in adulthood [90]. For instance, perinatal infection with hepatitis B virus results in persistent infection in approximately 90% of cases, whereas infection in adults only results in 5% of cases becoming persistent [90]. In the case of BTV, a study has found that infected newborn calves become PCR negative by 6 months [70]. BTV is therefore unlikely to produce chronic infections in young animals, but rather, as in the case of infection in adults [91], prolonged viremia is observed. This feature of BTV infection is thought to facilitate the transfer of the virus back to the vector and could possibly contribute to the re-emergence of the virus in spring.

The effects of BTV infection in early life on the repertoire of cells that respond to BTV are unknown. Infection could lead to an immunocompromised repertoire of T and B cells that respond to the virus. Further longitudinal studies will be required to assess the effects of BTV infection on adaptive immunity at different timepoints in animals’ lives. A compromised adaptive response to BTV due to an early life encounter with the virus could contribute to the characteristic prolonged viremia as adaptive immune cells fail to be optimally activated upon subsequent encounters. Indeed, we have shown in sheep that BTV limits humoral responses by targeting follicular dendritic cells, and this delays antibody response and potentially reduces IgG affinity for BTV antigens [92]. Furthermore, BTV infection is known to produce leukopenia [93] and, in some cases, limits the response to T cell mitogens [94]. These immunosuppressive phenomena could prolong virus circulation. Further work will be required to fully elucidate the effects of BTV infection on the immune system of young ruminants and determine whether infection in early life leads to deleterious effects on viral recognition in later life.

It is important to note that vertical transmission appears to be a feature of BTV infections limited to some strains. As previously mentioned, vertical transmission was initially thought to result from virus adaption to tissue culture conditions that favored the transmission through the transplacental barrier [63]. The overwhelming evidence that the field BTV-8 strain responsible for the 2006 European outbreak can be transmitted vertically has nonetheless challenged this view [44,45,46,67,68,69,70]. Since BTV genetic material is segmented, host co-infection with several BTV serotypes can result in reassorted viral progeny (i.e., a viral progeny in which segments that originate from the different serotypes are mixed) [95]. Sequence analysis indicated that the BTV-8 strain responsible for the outbreak in Northern Europe in 2006 did not originate directly from the BTV-8 live attenuated vaccine strain, but that it was a reassortant carrying segments from different serotypes [96]. This could have led to the introduction of the genetic determinants responsible for transplacental transmission in this strain. In the absence of studies that characterize the viral factors responsible for vertical transmission, it is difficult to discuss whether this feature was always present in the field strains or was introduced as a result of reassortment of field strains with live attenuated vaccine strains. Evidence that a reverse genetic BTV-2 strain was still capable of vertical transmission indicates that this characteristic is part of the virus make-up [44]. In any case, it is now clear that vertical transmission can be a feature of some BTV outbreaks and should therefore be monitored given its impact on reproduction.

## 5. Vaccination as a Strategy to Prevent BTV Vertical Transmission

Vaccination remains one of the most effective methods to combat infectious disease. This prophylaxis is probably the most cost-effective control method to prevent disease spreading: it protects animals, limits or stops disease transmission, and saves on resources that would have to be destined for disease treatment. Vaccination is an essential tool in animal health and in the fight against poverty [97].

Vaccination that would prevent BTV vertical transmission has several benefits for ruminant production. It would limit the abortions, stillbirths and weak offspring that result from in utero BTV infection, thus increasing productivity. It would also limit the possibility of disease overwintering in temperate climates, as newborns would not carry infective BTV, and thus could not trigger a new cycle of infection in the spring. Maternal vaccination could also provide passive immunity to the offspring through antibody transfer by colostrum intake after birth [98]. In the case of BTV, protection through colostrum intake could prevent newborns from becoming a reservoir for BTV transmission.

Vaccines are still being developed for arboviruses such as Chikungunya, dengue and Zika viruses, which can be transmitted vertically in humans [99]. An important consideration for these vaccines is their capacity to block vertical transmission, as these infections can have severe implications for the fetus [99]. The current vaccine for dengue virus is not recommended during pregnancy as insufficient data is available on its benefit [100], while preclinical studies have demonstrated some promising results for Zika virus candidate vaccines in preventing vertical transmission [101,102]. Evaluation of vaccine efficacy in terms of protection from vertical transmission in clinical trials can be difficult in these diseases owing to the unpredictable nature of arbovirus outbreaks. This implies that robust preclinical models are necessary to evaluate the effects of vaccination on vertical transmission.

Models to study BTV vertical transmission have been described in ruminants and mice [103,104]. Infection of pregnant ruminants in the most susceptible gestation period (typically between 1/3rd and 2/3rd of the gestation period) has been used to study the frequency of vertical transmission and BTV teratogenic effects [103]. A murine model in which the type I IFN receptor activity was blocked by antibody injection has also been described to study BTV transplacental transmission [104]. The classic IFNAR^(-/-)^ murine model for screening BTV vaccines [105] is, however, unlikely to be useful to study vaccine effectivity against vertical transmission as infected mice typically succumb to the disease within 5–10 days. Thus, the assessment of vaccination efficacy against transplacental transmission will require the use of BTV vertical transmission models.

The identification of BTV strains that are consistently capable of vertical transmission is also a requisite to study not only the pathogenesis of the infection but also the putative capacity of vaccines to prevent transmission through this route. Indeed, there is evidence that vaccination with inactivated BTV vaccines can limit vertical transmission of BTV-8 [71]. Santman-Berends et al. showed that none of the 256 calves born from BTV-8 vaccinated dams were positive by PCR for BTV [71]. Moreover, 13 dams that were seropositive before pregnancy did not give birth to BTV positive calves, indicating that exposure to the same BTV strain prior to pregnancy may also limit vertical transmission events [71]. There is a report of a calf born with hemorrhagic artery lesions (a hallmark of BTV infection) from a vaccinated dam, although the calf was negative by PCR at the time of assessment [106]. Overall, it appears that vaccination with inactivated vaccines could limit BTV vertical transmission, although further work will be required to confirm this. In the next section we will provide a brief overview of the vaccination strategies being developed for BTV and whether they could protect from vertical transmission.

## 6. BTV Vaccines: Live Attenuated, Inactivated or Recombinant Vaccines?

The pros and cons of BTV vaccine strategies are summarized in Table 1. As previously discussed, the main problem of BTV live attenuated vaccines is the possibility that, in spite of their attenuation, they acquire a phenotype capable of crossing the placental barrier that leads to abortions and teratogenesis in the fetus [61,77,78,79]. Moreover, live attenuated vaccines can be contaminated with exogenous viruses that can be pathogenic in some cases [72,107,108]. These drawbacks led to the development of inactivated BTV vaccines, which are effective and safe, but typically protect against only one serotype. The reduction in incidence of BTV-8 vertical transmission in vaccinated dams indicates that classical inactivated BTV vaccines can also offer protection to the fetus [71]. This is probably the result of the protection provided to the mother by the vaccine, which limits infection, and of antibody transfer from the mother to the newborn, which protects the newborn in early life. Overall, it appears that immunity to BTV can limit vertical transmission, but little is known on the mechanisms that afford this protection.

Another issue of “classical” vaccines is that they cannot differentiate infected from vaccinated animals (the so-called DIVA approach). A DIVA vaccine simplifies serological surveillance of vaccinated populations; this is therefore highly recommendable for disease control in disease-free regions that are at risk of outbreaks. DIVA vaccines are also ideal for eradication programs as they allow surveillance once vaccination campaigns are finished and animal trade is ready to resume. Typically, “classical” vaccines” only offer protection against re-infection with a virus from the same serotypes, which implies that multiple BTV vaccines need to be administered in regions where several serotypes are circulating. Thus, one of the ultimate goals in BTV vaccinology is to develop vaccine formulations that provide protection against multiple serotypes. Advances in molecular biology and recombinant protein technology have promoted the development of vaccine alternatives to BTV live attenuated and inactivated vaccines that aim to overcome these drawbacks of “classical” vaccines.

Broadly speaking, alternative BTV vaccines can be divided into three categories: (1) recombinant BTV protein vaccines; (2) live reverse genetics BTV vaccines; and (3) viral vector vaccines expressing BTV proteins [109]. The capacity of these vaccines to prevent vertical transmission has not been tested so far, but it is likely that if they confer good BTV immunity they will also limit all transmission routes. It should be noted that the description of a murine model of vertical transmission [104] could now allow testing of these alternative vaccine formulations in a preclinical model, thus facilitating the screening of candidate vaccines that could prevent vertical transmission.

Recombinant BTV protein vaccines include BTV subunit proteins expressed by different systems (insect cells [110,111], plant [112], yeast [113]); or bluetongue virus-like particles [114] that consist of the BTV capsid proteins expressed without the virus genetic material. Recombinant BTV protein vaccines can elicit immune responses in ruminants and even provide protection [115,116,117]. These approaches are deemed extremely safe, as these formulations are unable to replicate and therefore cause disease. They are also DIVA, as serological tests can easily differentiate animals vaccinated with vaccine subunits, as opposed to infected animals, which will also present antibodies to BTV proteins that are not present in the vaccine formulation. In spite of their safety, these approaches remain nonetheless quite expensive for veterinary medicine, and inactivated whole virus vaccines, which are cheaper to manufacture, are preferred. The advent of plant-based expression systems for these BTV constructs [112,115] could, however, change this in the long-term.

Reverse genetics technology for BTV [118] has opened new doors for the development of live vaccines in the field. This has allowed, for instance, for the introduction of alternative serotype-defining outer core proteins (VP2 and VP5) on the backbone of a live attenuated virus [119]. Disabled infectious single cycle (DISC) BTV vaccines have been developed by packaging a segment 9 that contains large deletions into the viral particle [120]. Since segment 9 encodes for the helicase VP6 that is critical for new viral particle assembly [24], this DISC virus could infect and express BTV RNA (except for VP6) but could not package new viral particles and thus spread in the host. Disabled infectious single animal (DISA) vaccines have also been described [121,122,123]. This was achieved by a deletion in segment 10 that encodes for NS3/NS3a. NS3/NS3a is not required for replication in mammalian cells but it is critical for virus release from *Culicoides* spp. [124]. Thus, the DISA vaccine can replicate in the ruminant host but cannot in the vector [124]. These live BTV vaccines designed by reverse genetics have been shown to protect ruminants from virulent virus challenge [119,120,121,122]. Because they mimic a natural infection, they have the potential to be effective as a single dose vaccine. Diagnostic tests can also be designed so that they can be considered DIVA vaccines. The risk of reversion to virulence is nonetheless still present, and reassortment during concomitant infection with wild-type BTV remains a possibility. Moreover, because attenuation has been associated with vertical transmission for some vaccine strains [61,63,65], the safety assessment of these live reverse genetics attenuated vaccines should probably include their capacity to cross the placental barrier.

Viral vector vaccines are based on the premise of activating innate immunity to provide sufficient adjuvancy so that an adaptive immune response is mounted to the antigen expressed by the vector [125,126,127]. Several platforms have been employed to induce immunity to BTV in the natural host. These include, among others, poxviruses [128,129,130], adenoviruses [130,131,132], Rift Valley fever virus (RVFV) [133,134], or herpesviruses [135,136]. These recombinant constructs were able to induce immunity to BTV in murine models and/or in the natural host, and protection was also demonstrated in some studies in the natural host [129,130,131,133,134]. It should be noted that most protection studies with viral vectors in ruminants only detected partial protection, as in most cases, in spite of the absence of clinical signs, some level of viral replication could be detected by PCR. Vaccines based on viral vectors nonetheless offer multiple advantages over “classical” vaccines. They are typically thermotolerant formulations, which facilitate transportation to remote areas with little infrastructures. They are non-pathogenic as they are often based on replication-defective viruses. They can offer protection over multiple BTV serotypes with the same formulation [128,132]. Recombinant vector vaccines based on attenuated vaccine strains, such as RVFV, can even induce bivalent protection in ruminants against RVFV and BTV [133]. They are also DIVA vaccines as only a fraction of BTV antigens are expressed in the recombinant vector, and thus a DIVA diagnostic test can be designed around these formulations. The correct cocktail of BTV antigens that provides a broad spectrum of protection is still the subject of active research in the field. It remains to be determined whether the immunity to BTV that viral vector vaccines produce is sufficient to limit vertical transmission.

**Table 1 pathogens-10-01528-t001:** Pros and cons of BTV vaccine strategies.

	Vaccine Type	Protection	Risk of BTV Vertical Transmission	DIVA ^1^
Classical	Live attenuated	Yes (serotype specific)	Possible	No
	Inactivated	Yes (serotype specific)	No	No
Alternative	Recombinant protein			
	BTV proteins [110,111,112,113]	Yes	No	Yes
	BTV VLP ^2^ [114]	Yes	No	Yes
	Live reverse genetics			
	DISC ^3^ [120]	Yes	Unlikely; Needs to be tested	Yes ^5^
	DISA ^4^ [121,123]	Yes	Needs to be tested	Yes ^5^
	Viral recombinant vectors			
	Poxvirus [128,129,130]	Yes ^6^ (potential for multiserotype)	No	Yes
	Adenovirus [130,131,132]	Yes ^6^ (potential for multiserotype)	No	Yes
	Rift Valley Fever Virus [133,134]	Yes ^6^ (bivalent BTV and RVFV)	No	Yes
	Herpesvirus [135,136]	Yes ^6^ (not tested in natural host)	No	Yes

^1^ DIVA: differentiation between infected and vaccinated animals. ^2^ VLP: virus-like particle. ^3^ DISC: disabled infectious single cycle. ^4^ DISA: disabled infectious single animal. ^5^: DIVA test needs to be designed around segment product deficiency. ^6^: Protection is often partial.

As previously stated, none of these experimental vaccines have been tested for their potency in inhibiting BTV vertical transmission. Data from vaccinated cattle with inactivated vaccines indicates that inducing good immunity to BTV is probably sufficient to greatly limit vertical transmission and therefore prevent abortions and newborn malformations [71]. It would therefore be interesting to evaluate whether these experimental vaccines can limit the transmission of BTV strains prone to cross the placental barrier.

## 7. Conclusions

Even though vertical transmission has long been associated with live attenuated BTV vaccine strains, the 2006 BTV-8 outbreak in Europe demonstrated that vertical transmission could be a feature of some BTV field strains. In utero infection can lead to abortions and/or congenital malformations that limit ruminant productivity. Moreover, vertical transmission can also contribute to the disease overwintering in temperate climates in which vector activity is reduced in colder months. As such, this transmission route and its consequences on reproduction should be monitored during BTV outbreaks. The factors involved in the crossing of the placental barrier by the virus remain elusive, and thus further work will be necessary to pinpoint these. As often seen, vaccination appears to be an effective tool to limit disease spreading and to impair the teratogenic effects of BTV. Establishing adequate models of BTV vertical transmission will also help in the development of strategies to counter this transmission route. Since classic models for screening BTV vaccine candidates are unlikely to be useful in protection studies against vertical transmission, establishing robust models of vertical transmission for BTV will be a necessity. This includes the characterization of BTV strains prone to transmission through this route as well as precisely defining the experimental conditions that favor transplacental barrier crossing. These issues are critical to adequately assess vaccine efficacy against vertical transmission. Much work remains to be done to fully understand BTV capacity to be transmitted vertically and produce harm to the developing fetus.

## Figures and Tables

**Figure 1 pathogens-10-01528-f001:**
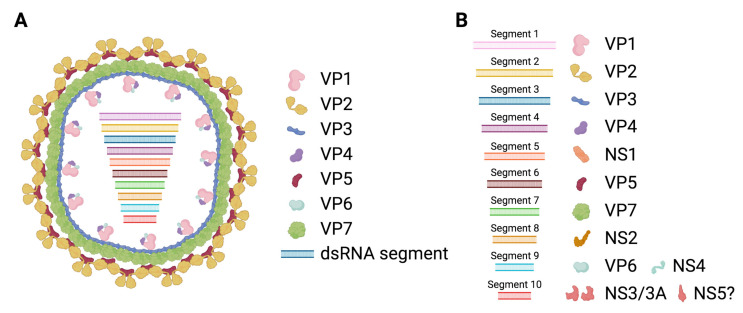
Schematic representation of bluetongue virus (BTV). (**A**) The bluetongue viral particle is composed of an outer capsid that consists of the VP2 and VP5 proteins, and an inner core formed by the VP7 and VP3 proteins. VP3 anchors the RNA polymerase VP1 to the capsid. The RNA capping and methyl transferase VP4 and the helicase VP6 are associated with VP1. Enclosed within the inner core, the BTV genome consisting of 10 segments of dsRNA is found. (**B**) The segmented genome of BTV encodes for 7 structural proteins (VP1 to VP7) and at least 4 non-structural proteins (NS1 to NS4). Segment 1 encodes for the RNA polymerase VP1. Segment 2 encodes for the highly variable VP2. Segment 3 encodes for the inner core protein VP3. Segment 4 encodes for the methyl transferase and RNA capping enzyme VP4. Segment 5 encodes for NS1, a non-structural protein that forms cytoplasmic tubules. Segment 6 encodes for the outer capsid protein VP5. Segment 7 encodes for the inner core protein VP7. Segment 8 encodes for NS2, an RNA binding non-structural protein expressed in viral inclusion bodies. Segment 9 encodes for the helicase VP6 and for NS4, a non-structural protein involved in immune evasion. Segment 10 encodes for NS3 and its isoform NS3a, which are polyfunctional non-structural proteins involved in viral particle exit from the cell as well as in interference with the mammalian IFN system. Segment 10 also putatively encodes for a fifth non-structural protein (NS5), which could be implicated in cellular shutdown. (Created with Biorender.com).

## Data Availability

Not applicable.

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
