# Peer review of "Vaccination as a Strategy to Prevent Bluetongue Virus Vertical Transmission"

_pathogens, 2021, doi:10.3390/pathogens10111528_

Round 1
Reviewer 1 Report
The paper "Vaccination strategies to prevent bluetongue virus vertical 2 transmission" adequately summarizes the vaccination strategies that can avoid vertical transmission of BTV in animals. Among the possible undesirable effects of the use of live attenuated vaccines I would also suggest the possible introduction of other exotic viruses. The almost simultaneous appearance in Europe of BTV8 and Schmallenberg viruses, both causing abortions and fetal malformations in different animal species and both transmitted by Culicoides spp, could suggest that these two viruses, of African origin, have arrived via the same route. The pathogenetic characteristics of BTV8 2006 also suggest its origin from a vaccine strain.
Author Response
We wish to thank reviewer 1 for his/her kind comments. Please find below a reply point by point to his/her comments.
"The paper "Vaccination strategies to prevent bluetongue virus vertical 2 transmission" adequately summarizes the vaccination strategies that can avoid vertical transmission of BTV in animals.
Among the possible undesirable effects of the use of live attenuated vaccines I would also suggest the possible introduction of other exotic viruses."
We have now included and referenced this drawback of live attenuated vaccines in the manuscript.
"The almost simultaneous appearance in Europe of BTV8 and Schmallenberg viruses, both causing abortions and fetal malformations in different animal species and both transmitted by Culicoides spp, could suggest that these two viruses, of African origin, have arrived via the same route."
This an interesting point, although we think that discussing the route of entry of BTV-8 into Europe is beyond the scope of the present review.
"The pathogenetic characteristics of BTV8 2006 also suggest its origin from a vaccine strain."
Sequence analysis by Maan et al in 2008 (Maan et al, 2008, Virology. 377:308-318) indicated that BTV-8 NET2006 did not originate directly from the vaccine strain for BTV-8. Nonetheless, they also provide evidence that BTV-8NET2006 is the result of reassortment between serotypes, and thus it cannot be excluded that reassorting events may have led to the introduction of the genetic determinants responsible for transplacental transmission that originated from a vaccine strain. This is now mentioned in the manuscript.
Reviewer 2 Report
This is a nice review on Blue Tongue virus biology and vaccines that could prevent vertical transmission.
The review is well written although it has some minor grammar mistakes worth to check out.
Perhaps the title is a bit deceiving in the sense that it focuses on vaccines but over 50% of the content goes over the biology of the virus, and this is OK and very well summarized. Perhaps reorganizing the words in the title would better summarize the content of the review.
Figures are nice. In my opinion inclusion of a table summarizing the available vaccines with pros and cons would strengthen the good quality of the manuscript.
Minor English mistakes
Line 24: World Organization of Animal Health
Line 33: Consists of
Line 38: Mostly instead of more particularly
Line 43: indicate which are the classical BTV serotypes
Line 62: “protective shell for the viral replication machinery” (?)
Line 67: RNA genome
Line 83: NS5 has been shown…
Author Response
We wish to thank reviewer 2 for his/her kind comments. Please find below a reply point by point to his/her comments.
"This is a nice review on Blue Tongue virus biology and vaccines that could prevent vertical transmission.
The review is well written although it has some minor grammar mistakes worth to check out.
Perhaps the title is a bit deceiving in the sense that it focuses on vaccines but over 50% of the content goes over the biology of the virus, and this is OK and very well summarized. Perhaps reorganizing the words in the title would better summarize the content of the review."
The title has been changed to addresss this.
"Figures are nice. In my opinion inclusion of a table summarizing the available vaccines with pros and cons would strengthen the good quality of the manuscript."
As suggested, we have now included a table summarizing the pros and cons of the different vaccination strategies developed for BTV.
"Minor English mistakes
Line 24: World Organization of Animal Health
Line 33: Consists of
Line 38: Mostly instead of more particularly
Line 43: indicate which are the classical BTV serotypes
Line 62: “protective shell for the viral replication machinery” (?)
Line 67: RNA genome
Line 83: NS5 has been shown…"
These mistakes have been corrected. The manuscript has been rechecked for mistakes